# Physical Activity and Mental Health of Medical Students from Poland and Belarus-Countries with Different Restrictive Approaches during the COVID-19 Pandemic

**DOI:** 10.3390/ijerph192113994

**Published:** 2022-10-27

**Authors:** Joanna Baj-Korpak, Kamil Zaworski, Ewa Szymczuk, Andrei Shpakou

**Affiliations:** 1Department of Physiotherapy, Faculty of Health Sciences, John Paul II University of Applied Sciences in Biala Podlaska, 21-500 Biala Podlaska, Poland; 2Department of Nursing, Faculty of Health Sciences, John Paul II University of Applied Sciences in Biala Podlaska, 21-500 Biala Podlaska, Poland; 3Department of Theory of Physical Culture and Sports Medicine, Yanka Kupala State University of Grodno, 230023 Grodno, Belarus; 4Department of Integrated Medical Care, Faculty of Health Sciences, Medical University of Bialystok, 15-089 Bialystok, Poland

**Keywords:** COVID-19, physical activity, mental health, well-being, medical students

## Abstract

Background: COVID-19 pandemic has struck all of us suddenly and unexpectedly; it deprived the society of a sense of control over their lives on different levels. In a short period of time, it led to a number of changes in everyday life of people all over the world. In particular, these changes affected medical staff, who, all of a sudden, were burdened with new work-related responsibilities and duties. This situation may have had a detrimental effect on their mental health. Due to the unpredictability of the COVID-19 pandemic, we attempted to assess its consequences in terms of mental health and physical fitness of university students from countries in which different approaches to these issues were adopted. Methods: A total of 779 medical students (374 students from John Paul II University of Applied Sciences (ABNS) in Biala Podlaska, Poland, and 405 students from Yanka Kupala State University of Grodno (YKSUG), Belarus) took part in the survey. Three standardised psychometric tools were used in the study: The Satisfaction With Life Scale (SWLS), The General Health Questionnaire (GHQ-28) and Stress Coping Inventory (Mini-COPE). In addition, the International Physical Activity Questionnaire (IPAQ) was applied. Results: The vast majority of students both from Poland and Belarus demonstrated high levels of physical activity. However, students from ABNS manifested significantly higher levels of physical activity compared to their counterparts from YKSUG. Students from Biala Podlaska had greater satisfaction with life during the COVID-19 pandemic, whereas their peers from Grodno exhibited higher levels of mental distress. Conclusion: The COVID-19 pandemic resulted in a significant exacerbation of mental health issues among medical students. In order to alleviate negative effects of the pandemic, it seems necessary for universities to monitor the physical and mental health state of students and to implement prevention programmes.

## 1. Introduction

Coronaviruses are a large family of pathogens which are phenotypically and genotypically diverse. COVID-19 is a highly infectious respiratory disease caused by severe acute respiratory syndrome coronavirus-2 (SARS-CoV-2) [1].

The first COVID-19 outbreak occurred in Wuhan, China, in December 2019. Afterwards, it rapidly spread around the world [2]. The first confirmed case of COVID-19 in Poland was noted on 4 March 2020. The Polish government declared a state of emergency due to the pandemic on 20 March 2020. It resulted in numerous restrictions related to movement, access to the most important public services such as childcare, education or medical services [3]. For a long time, no anti-pandemic measures were taken in Belarus. Although the first case of coronavirus infection was reported there as early as 27 February 2020, the threat was consistently downplayed. During the first wave of the pandemic, no decision was made to close the borders. In the first months of the pandemic, neither schools nor shops were closed and no sporting events were suspended [4]. To date, over 550 million cases of SARS-CoV-2 infection and more than 6 million disease-related deaths have been confirmed. In Poland, over 6 million people have been infected and 116,000 have died because of the disease. In Belarus, 983,000 COVID-19 cases have been confirmed and 6987 people have died [5].

One of the main restrictions imposed to reduce the spread of the disease was the introduction of quarantine, which meant limiting contact with other people including family and friends. Based on the literature review, Brooks et al. [6] pointed to a lot of negative psychological consequences of quarantine such as confusion, anger or post-traumatic stress disorder as well as increased occurrence of anxiety and depression in the general population. In the long term, quarantine and self-isolation increase stress related to fear of being infected or insufficient care. According to these researchers, the most common psychological and behavioural reactions during the quarantine include irritability, nervousness, frustration, emotional distress, sadness, a sense of guilt, exhaustion, boredom, insomnia, the lack of information, poor concentration, indecisiveness, alienation, poorer job-related performance, financial problems and censure.

The pandemic has exerted a negative influence on the quality of social life of university students. Being a student involves not only studying but also establishing new relationships and experiencing intense social life. Research indicates that contact with others has a positive impact on life quality [7]. Not being able to spend time with friends results in loneliness, which may not be fully compensated for through regular phone calls or other forms of contact [8]. Increased occurrence of depression and anxiety both during and after periods of social isolation was confirmed, which, according to Van Lancker [9], may have negative social and psychological consequences. Poor stress management over a longer period of time leads to emotional and psychosomatic consequences manifested through physical, cognitive and emotional exhaustion as well as reduced learning effectiveness [10].

Students constitute the population particularly prone to mental disorders in the context of challenges associated with becoming an independent adult as well as coping with financial issues [11,12].

Society was not prepared for unexpected changes, and e-learning proved to be highly stressful for young people. Distance education caused a lot of difficulties both for students and academic staff [13]. 

Research carried out so far shows that a lot of students struggle with numerous stress-related issues, and the prevalence of stress in this group is constantly increasing. Chronic stress has a detrimental effect on mental health [14,15]. Beiter et al. stated that stress is a part of students’ lives, as their everyday duties pose numerous challenges that enhance its experience [16].

The COVID-19 pandemic affected everyone, including students. Those who did not get infected also had to comply with different forms of restrictions introduced by a lot of countries that aimed to prevent the spread of the disease [17].

Well-being is considered to be one of the indicators of health, as it is often associated with it. According to Shin and Johnson [18], satisfaction with life is understood as overall assessment of life quality with reference to the criteria that we have selected. Satisfaction with life is defined as an individual’s perception of their life situation in the context of culture, objectives, expectations and fears [19]. Moreover, it is closely related to mental health [20].

The level of knowledge regarding medical, psychological and socio-cultural determinants of physical activity and mental health still seems to be unsatisfactory.

The present study sought to determine PA levels, satisfaction with life and mental health indicators in students from Poland and Belarus in the context of different approaches to the COVID-19 pandemic.

Given that the COVID-19 pandemic affected all of us unexpectedly, the study aimed to assess its consequences in terms of mental health status and PA levels among students from countries in which different approaches to these issues were adopted. Similarly to other countries in the European Union, Poland introduced severe quarantine measures (lockdown) in order to control the spread of the disease. The restrictions imposed in Belarus in connection with the COVID-19 pandemic turned out to be much less pronounced in comparison with anti-pandemic measures in Poland. Belarus is one of the few countries in Eastern Europe where no lockdowns have occurred. Belarus did not endorse quarantine and proceeded with “life as usual”.

## 2. Materials and Methods

### 2.1. Participants

A total of 779 medical students (374 students from John Paul II University of Applied Sciences (ABNS) in Biala Podlaska, Poland, and 405 students from Yanka Kupala State University of Grodno (YKSUG), Belarus) took part in the survey. A purposive sampling method was applied; it was assumed that all medical students from ABNS and 100 students from each year (years 2 to 5) from YKSUG would be included in the survey. The universities are situated not far away from each other. However, the differences concerned the approaches that both countries adopted towards the pandemic. The characteristics of the study groups can be found in Table 1. 

Taking into account the sex of the study participants, no significant correlations were noted between students from ABNS and YKSUG. As it has already been mentioned, students from Grodno were younger than their peers from Biala Podlaska (the correlation between variables was strong and statistically significant). As for the place of residence, more participants from YKSUG came from towns and cities compared to students from ABNS (the correlation coefficient was strong and statistically significant). Self-isolation during the COVID-19 pandemic occurred significantly more often among students from Grodno (however, the correlation was weak). Respondents from ABNS were quarantined significantly more often than their peers from YKSUG (the strength of correlation was inconsiderable). More students from ABNS declared that they were vaccinated against COVID-19 compared to their counterparts from YKSUG. What is more, they stated that they received more doses (the correlation between variables was strong and statistically significant). Participants from Grodno reported that they were diagnosed with SARS-CoV-2 more frequently than those from Biala Podlaska (the correlation coefficient was statistically significant but the correlation was weak) (Table 2).

### 2.2. Methods

The auditorium face-to-face questionnaire was used in our study. The questionnaire was completed by the respondents at their place of study, with one of the researchers present. The study was carried out in April and May 2022. The results presented in the current study had been obtained with the use of four questionnaires: The International Physical Activity Questionnaire (IPAQ)—the questionnaire consists of seven questions regarding types of physical activity that constitute elements of everyday life. The questions refer to physical activity in the last seven days and concern the time spent sitting and walking as well as the time devoted to vigorous and moderate physical activity. Activities done for at least 10 min at a time are taken into account. This instrument is used to assess physical activity levels—according to the IPAQ scoring protocol, three levels are distinguished, i.e., low, moderate and high [21];The Satisfaction With Life Scale (SWLS) developed by Diener et al. [22] from the Department of Psychology, University of Illinois—the scale includes five statements that an individual relates to by indicating how much they agree or disagree with them in terms of their life. The score is an overall indicator of satisfaction with life. The satisfaction with life measured with SWLS is manifested through a sense of satisfaction with one’s own achievements and conditions [23]. The questionnaire is designed for adults (both healthy and ill ones). It is a useful instrument for measuring life satisfaction per se [24]. Cronbach’s (α) reliability analysis was used in order to verify the internal consistency of the questionnaire: the reliability of the tool was assessed as α = 0.872, which is a satisfactory level of reliability;Mental health was assessed using the General Health Questionnaire (GHQ-28) developed by David Goldberg [25]. It refers to changes in a subjective perception of health whose possible deterioration may stem from experiencing problems, difficulties, worries or mental illness. The questionnaire also measures the improvement in mental health state resulting from the influence of different environmental factors. It is used to assess environmental determinants of mental health state, which means it is particularly useful when pursuing the aim of the present study in the context of the COVID-19 pandemic. GHQ-28 measures four aspects of mental health: A-somatic symptoms, B-anxiety and insomnia, C-social dysfunction, D-depression symptoms. In the present study, the Cronbach’s alpha coefficient for GHQ-28 was 0.95;Mini-COPE-Stress Coping Inventory developed by Charles S. Carver is a tool for adults (both healthy and ill ones). It consists of 28 statements that make up 14 strategies (2 statements in each strategy). The tool is used to assess typical ways of reacting to situations of severe stress. The strategies are divided into 4 categories (integral strategies) and corresponding scales: active coping (active coping, planning, positive revaluation), helplessness (taking psychoactive substances, doing nothing, self-accusation), seeking support (seeking emotional and instrumental support), avoidance behaviours (dealing with other things, denial, giving vent to one’s feelings). Such strategies as turning to religion, acceptance and sense of humor constitute separate categories [26,27]. In the present study, the Cronbach’s alpha coefficient for Mini-COPE was 0.789.

This study was carried out within the project Physical activity and mental health of medical students from Poland and Belarus in the context of the dynamically changing situation of the COVID-19 pandemic funded by the National Agency for Academic Exchange (NAWA). The research tools (psychological tests) were purchased from the Psychological Test Laboratory of the Polish Psychological Association. The usefulness of the tests had been discussed by the researchers before [28].

The study was conducted in accordance with the Declaration of Helsinki and the Bioethics Committee of the ABNS in Biala Podlaska (Resolution no. 4/2022) approved the protocol.

### 2.3. Statistical Analysis

Research results were analysed using SPSS 17.0 (Softonic, Ashburn, VA, USA). To calculate qualitative data, two correlation coefficients based on the Chi-squared test were used, i.e., Phi and V Kramer.

Quantitative data were presented taking into account such descriptive characteristics as mean (M) and standard deviation (SD). Comparative analysis between groups ABNS and YKSUG was carried out using the T test for independent samples. If high SD values were observed, U Mann–Whitney non-parametric test was applied. Effect size was determined using Cohen’ d, Hedges’ g, and Glass’s delta. The following Cohen’s criteria were adopted: 0.2 small effect, 0.5 medium effect, and 0.8 and above-large effect. Correlations between qualitative variables were calculated using Spearman’s rho, which measures the strength and direction of relationships between variables. Statistical significance was set at *p* < 0.05.

## 3. Results

Given the aim of the study, we began our analyses with determining PA levels (according to IPAQ methodology) demonstrated by university students from Biala Podlaska and Grodno during the COVID-19 pandemic (Table 3). In both groups, the percentage of students with high PA levels was similar. Moderate PA levels were noted more often in students from YKSUG, whereas low PA levels were mainly found in students from ABNS. The correlation between variables was statistically significant, while its strength proved to be low.

Comparative analysis of variables between groups ABNS and YKSUG was performed using the T test for independent samples (Table 4). Significant differences between the results obtained in Poland and in Belarus were found in SWLS. Greater satisfaction with life during the COVID-19 pandemic was observed in students from Biala Podlaska.

The analysis of strategies of coping with stress measured with Mini-COPE revealed that the students under study differed significantly in terms of coping strategies. Students from ABNS used strategies based on active coping, planning, acceptance and dealing with other things more frequently compared to their peers from YKSUG. Simultaneously, they made use of such strategies as taking psychoactive substances, denial and doing nothing less often than students from YKSUG.

The analysis of mental health state revealed higher values of indices in the group of students from Belarus in all of the four scales. Significant differences were noted in somatic symptoms, anxiety, insomnia and depression symptoms. 

The comparison between students from Poland and Belarus showed that both groups differed significantly in mental health state. Students from Grodno manifested higher levels of mental health issues. Students from both ABNS and YKSUG displayed higher levels of MET-min/week. Significantly higher levels of PA were noted in students from Poland (Table 4).

The analysis with the T test for independent samples revealed significant differences between the groups in MET-min/week. Students from ABNS manifested higher levels of PA than their counterparts from YKSUG. Due to the fact that standard deviations in both groups were considerable and not comparable, U Mann–Whitney non-parametric test was applied. It confirmed that the differences were statistically significant (Table 5).

The analyses showed that in ABNS, there was a significant moderate correlation between life satisfaction and negative symptoms of depression and a weak correlation between life satisfaction and negative somatic symptoms, anxiety and insomnia as well as social dysfunction. Furthermore, students from ABNS exhibited significant weak correlations between PA and somatic symptoms. The higher their satisfaction with life, the fewer health disorders such as somatic symptoms, anxiety and insomnia or depression symptoms they experienced. In addition, the more physically active they were, the fewer somatic symptoms they had. The analyses did not reveal any correlations between their PA levels and mental disorders. In YKSUG, significant weak correlations were observed between life satisfaction and health disorders manifested through somatic symptoms, anxiety and insomnia, social dysfunction and depression symptoms. The more satisfied they were with their lives, the greater their health deterioration in the aspects mentioned below. There was, however, a weak correlation here (Table 6).

The analysis of SWLS results in the context of IPAQ scores did not show any significant correlations between satisfaction with life and PA levels in students from both ABNS and YKSUG (Table 7).

Table 8 shows correlations between coping with stress and physical activity, mental health and life satisfaction during the pandemic. In the case of strategies of coping with stress, the table includes both general factors that were highlighted by the tool developers as well as detailed strategies of coping with stress.

In the case of the study participants from ABNS, a significant moderate correlation was found between how they made use of active coping strategies and mental health disorders in the aspect of depression. Moreover, weak correlations were noted between active coping strategies and PA levels, somatic symptoms and social dysfunction. The more these students used active coping strategies, the less they experienced negative depression symptoms, somatic symptoms and social dysfunction. In addition, they manifested greater satisfaction with life and higher PA levels. 

The participants from ABNS used active coping strategies, planning and seeking emotional support significantly more often when they obtained higher scores on SWLS and lower depression-related scores on GHQ. Conversely, lower scores on SWLS and higher levels of depression symptoms correlated with a more frequent use of such strategies as doing nothing and self-accusation. As far as denial is concerned, it was used by those students from ABNS who exhibited higher levels of somatic symptoms, anxiety and insomnia (GHQ).

The analysis of general factors of Mini-COPE in students from Poland revealed moderate correlations between higher levels of mental disorders and higher levels of helplessness, and weak correlations between higher levels of mental disorders and higher levels of avoidance behaviours and lower levels of active coping. 

As for detailed strategies of Mini-COPE, the higher the levels of mental health disorders in students from ABNS, the more frequent the significant moderate correlations with such coping strategies as self-accusation, doing nothing, taking psychoactive substances, giving vent to one’s feelings and denial. In turn, they used strategies focused on active coping, planning, positive revaluation and sense of humor less often. 

The participants from YKSUG who manifested higher levels of depression symptoms (GHQ) used active coping strategies significantly less often. Those who turned to religion displayed higher levels of somatic symptoms, anxiety and insomnia (GHQ). Students from Grodno used such strategies as dealing with other things, denial, doing nothing, self-accusation and giving vent to one’s feelings significantly more often when they demonstrated higher levels of life satisfaction and obtained higher scores on all GHQ scales. Taking psychoactive substances was also linked with higher levels of general health. 

The analysis of general factors of Mini-COPE in students from Belarus showed that higher levels of mental health state correlated moderately with higher levels of helplessness and correlated weakly with higher levels of avoidance behaviours and lower levels of active coping.

Furthermore, it was noted that the higher the levels of mental health issues in students from YKSUG, the more frequent the significant moderate correlations with such coping strategies as self-accusation and doing nothing as well as strategies like turning to religion, taking psychoactive substances, giving vent to one’s feelings, denial and dealing with other things. In such cases, they used active coping strategies less often. 

Taking integral strategies into consideration, active coping and seeking support were employed by students from ABNS significantly more often when they obtained higher scores on SWLS and displayed lower levels of depression symptoms (GHQ). Helplessness was connected with lower satisfaction with life (SWLS) and higher levels of all types of symptoms on GHQ scales. Students from ABNS used the strategy of avoidance behaviours more frequently when they showed higher levels of somatic symptoms, anxiety and insomnia. In students from YKSUG, integral strategies that include active coping were used significantly more often by individuals with lower levels of functional dysfunction and depression symptoms. Strategies based on helplessness and avoidance behaviours were typical of students with higher levels of satisfaction with life (SWLS) and all GHQ symptoms.

## 4. Discussion

In the course of determining PA levels, satisfaction with life and mental health state of students from Poland and Belarus in the context of different approaches to the pandemic in the dynamically changing situation of the COVID-19 pandemic it was noted that the vast majority of students from Poland and Belarus manifested high levels of PA, thus meeting the criteria established by the World Health Organization (WHO) [29]. Students from ABNS exhibited significantly higher levels of PA in comparison to their peers from YKSUG. The results of Rousset et al. [30] also indicate that young adult students from France followed PA recommendations put forward by WHO despite pandemic restrictions. In addition, the findings of the current study revealed higher levels of life satisfaction during the COVID-19 pandemic in students from Biala Podlaska. In turn, students from Grodno displayed higher levels of mental health state indices.

Changes in mental health state seem to have become an element characteristic of the COVID-19 pandemic. Studies show that approximately a third of the Chinese population suffered from various forms of mild to severe depression and anxiety due to the lockdown following the outbreak of the COVID-19 pandemic [31,32,33]. Women manifested lower levels of satisfaction with life and higher levels of stress and anxiety [33]. Individuals aged 21 to 40 were more susceptible to mental disorders and alcohol abuse than those from other age ranges [34]. In their meta-analysis of longitudinal studies carried out during the pandemic, Prati et al. [35] mainly point to negative changes concerning depression and anxiety. According to Fond et al. [36], it is deeply disturbing because the risk of death among patients with mental disorders is greater, and a higher prevalence of such disorders is correlated with a reduced occurrence of health-oriented behaviours [37] such as taking up physical activity, which is in line with our findings. 

Deschasaux-Tanguy et al. [38] claim that during the pandemic it is particularly difficult to exhibit health-oriented behaviours including physical activity performance. What may facilitate the comprehension of the nature of this issue and help to counter it is the research results that explain why people do not do exercise during the pandemic and what motivates those who are physically active. According to Villadsen et al. [37], the main causes of physical inactivity were excessive anxiety, the lack of social support as well as the lack of access to sports facilities and equipment. Therefore, it is essential that the benefits of taking up physical activity manifested through improved mental health state be emphasised. If anxiety constitutes a barrier to performing physical activity, it is worth advertising such activity as a means of addressing this issue. What is more, Marashi et al. [39] noted that during the pandemic, physically active individuals are mainly motivated by the fact that physical activity contributes to improving mental well-being and reducing anxiety. The findings of the study on benefits of physical activity during the COVID-19 pandemic show that encouraging people to be physically active may lead to an improvement in their mental health state [40].

The results of the investigation conducted by Alsharji [41] confirm a positive correlation of PA levels with a reduction in anxiety and depression during the pandemic. Alleviating effects of PA on negative emotions experienced by Chinese students during the pandemic were also observed by Zhang et al. [32]. In their study on 1491 adult Australians, Stanton et al. [42] noted negative changes in PA, sleep as well as tobacco and alcohol use, which were associated with higher depression, anxiety and stress symptoms. 

Physically active individuals demonstrate lower levels of stress and better emotional state [43]. The more active our study participants were, the fewer health disorders manifested through somatic symptoms they experienced. Wolf et al. [40] proved that individuals who performed PA regularly displayed fewer mental symptoms. More active persons had 12% to 32% lower chances of presenting symptoms of depression and 15% to 34% lower chances of displaying anxiety. In addition, Coughenour et al. [44] found a significant correlation between PA and depression symptoms. However, the present study did not reveal significant correlations between PA levels and depression symptoms, anxiety and insomnia.

In the study of Coyle et al. [45], PA proved to be the most common activity performed by the respondents (students) in order to improve mental well-being (80.1%). Physically active students had higher mood scores than their non-active counterparts. Physical activity exerts a positive influence on students’ well-being. The findings of our study show that despite various restrictions introduced during the COVID-19 pandemic in Poland and Belarus, PA levels of the respondents from both countries were high, as indicated by MET-min/week values.

Shpakou et al. [46] noted a positive correlation of PA (IPAQ) with life satisfaction (SWLS). However, our results did not confirm such a correlation. The above-mentioned study of Shpakou et al. [46] showed that the majority of students from Belarusian universities demonstrated high PA levels. A considerable proportion of the respondents used such coping strategies as active coping, planning, seeking emotional support as well as positive revaluation and development. The researchers recommend monitoring mental and physical health as well as developing a common strategy of engaging students in broadly understood physical activity in order to cope with stress in the dynamically changing situation of the COVID-19 pandemic.

As early as in the first weeks after the COVID-19 outbreak in China, studies on mental health were carried out. They showed that the epidemic has a significant impact on mental health [47,48], and students are particularly susceptible to its deterioration. Liang et al. [49] point out that students experience mental health problems and have post-traumatic stress disorder symptoms.

In the current study, most respondents from Poland and Belarus were women. Women are more prone to stress and such disorders as anxiety and depression [50,51]. It was confirmed in the study on medical students: women obtained higher scores in terms of emotional exhaustion. Furthermore, they manifested reduced perceptions of physical and psychological quality of life (QoL) as well as increased inclination towards emphatic concern and feelings of distress and anxiety [52]. Bertrand et al. [53] claim that women exhibit higher levels of PA during the pandemic. They conclude that women are more motivated due to such factors as peer pressure or body mass loss.

Stress perception is a subjective and variable phenomenon. Particular attention is paid to processes of coping with stress, which determine its positive or negative influence on an individual. The course of a coping process depends on personal resources and social support, while styles of coping with stress are conditioned, inter alia, by sex, education, age, health state, well-being, personality or type of a stressful situation [54].

BabickaWirkus et al. [54] reported that the dominant coping strategies in Polish students were acceptance, planning and seeking emotional support, whereas the least common strategies were taking substances, denial, avoidance behaviours and turning to religion. The current study presents similar results. The strategies Polish and Belarusian students used most often were active coping and planning, while the ones applied the least frequently included self-accusation and taking psychoactive substances. Vanderbruggen et al. [55] revealed a considerable increase in alcohol and tobacco use among Belgian respondents compared to pre-pandemic levels. In turn, no significant changes in cannabis consumption were noted. It was mainly younger participants who used these substances more frequently. The main reasons for their consumption were boredom, the lack of social contacts, loneliness, the loss of daily structure and a need for reward after a hard working day. Our results indicate that students from Belarus used psychoactive substances as a way of coping with stress significantly more often than their peers from Poland.

We feel comfortable and safe when, based on the information we receive, we are capable of recognising, predicting and, to a certain extent, controlling the surrounding reality [56]. The COVID-19 pandemic disrupted relative predictability of the world and a sense of control over our lives in many areas. The majority of us, younger generation in particular, had never experienced anything closely resembling it, so, as is the case of any change, it requires gradual adaptation [57]. Individuals with decreased mental well-being experience greater stress and mainly make use of such strategies as denial, giving vent to one’s feelings, taking psychoactive substances, doing nothing and self-accusation, which is in line with our findings. It may be assumed that such persons consider the COVID-19 pandemic to be an inevitable but relatively remote danger, thus resorting to passive coping strategies [57].

In the present study, participants from Poland demonstrated higher levels of life satisfaction and lower values of somatic and mental symptoms (GHQ). An interesting phenomenon is a negative correlation noted in the group of the respondents from Belarus. It may point to considerable mental burden in students from YKSUG and it is consistent with the results of previous studies [58,59]. A substantial increase in mental disorders may have been caused by social pressure aimed to combat COVID-19 and a serious strain on medical staff including the medical students under study.

A deterioration in mental well-being is one of the major issues linked to the COVID-19 pandemic among students. Stress and depression may exert a profound influence on general health state in the years to come; therefore, it is necessary to manage people’s mental health properly, irrespective of age. Activities aimed at promoting physical activity and broadly understood healthy lifestyle may prove to be significant. Students should receive support in dealing with a new situation, e.g., by getting access to educational programmes and materials regarding prevention of mental health issues. Universities should offer access to specialists like psychologists who provide direct help to students in difficult situations. Taking care of young people’s mental health is an investment in the future [60]. The Aristotle University of Thessaloniki (Greece) has established a 24 h communication line for members who seek counselling and psychological support. Records so far have been in alignment with the evidence presented in this study. The cases of the University community members who seek consultation and psychological support have quintupled during the last two years. Therefore, it is necessary for universities to monitor physical and mental health state [61].

Given an uncertain course of COVID-19 in the months and years to come, our research results point to the need for an intervention aimed at instilling a habit of healthy lifestyle in younger generations that would enable them to maintain their health and cope with stress. It seems important from the point of view of broadly understood public health and health education, since such information is crucial when it comes to fulfilling the needs of young people concerning physical and mental health during the COVID-19 pandemic. 

Our findings may constitute a reference point for other studies on coping strategies among students from different countries. This, in turn, may influence the development of local strategies of supporting students in their educational careers and personal lives. 

### Limitations

The findings of our study have some limitations. The lack of representative national data from before the pandemic poses a problem when it comes to making comparisons. We recognise it and stress that caution should be exercised when comparing our results with previous data. We analysed the data related to the dynamically changing pandemic situation, but we could not make any deductions about the return to normal life because of the lack of data collected before the pandemic.

In our estimation, the strength of our study is the population that is representative of the general model of mental health of students from two different countries. The data presented reflect mental problems as well as the need for intervention in a particular social group during the COVID-19 pandemic. 

The female representation in the study sample was considerable. In both groups, sex proportions were similar. Sex is a significant factor determining PA: men usually demonstrate its higher levels. Moreover, they are normally less prone to negative emotions. Given the aforementioned limitations, correlations between PA during the COVID-19 pandemic and mental health may differ between women and men. In addition, preferred methods of coping with negative emotions such as stress, anxiety and depression are different in women and men.

The cities where the study was carried out are located on both sides of the state border between Belarus and Poland. The pandemic waves did not differ much in terms of their timing in these regions. Moreover, the borders between the countries remained open for a long time. For this reason, we considered the dynamics of the pandemic in these regions to be similar. In addition, the study did not focus on the incidence of the disease due to the fact that, unfortunately, official statistics tend to be inaccurate.

As the manuscript shows, the survey was carried out in April and May 2022. In Poland, the COVID-19 epidemic state was lifted by the Ministerial Decree of 13 May 2022. Due to a significant decrease in infections that occurred at that time as well as because the epidemic state was lifted, no specific dates (time) of peak infections (waves) were included in the analysis. The aim was to identify the effects of the pandemic manifested through PA levels and general mental health. In other words, the goal was to find out how students from Poland and students from Belarus felt afterwards (sense of life satisfaction, mental health indicators, physical activity levels).

## 5. Conclusions

Despite the fact that different approaches to the pandemic were adopted in Poland and in Belarus, the vast majority of the respondents manifested high levels of PA, which may be evidence of their high awareness of positive effects of physical activity on general health.

Significantly higher life satisfaction that is conducive to greater activity and better handling of difficult situations was noted in students in ABNS, while higher indices of mental health issues, both in general terms and in the case of somatic symptoms, anxiety, depression, insomnia and social dysfunction, were observed in students from YKSUG.

It seems important for universities to monitor physical and mental health state of students and to implement prevention programmes. 

Taking into account the changing situation of the COVID-19 pandemic, our findings constitute the basis for further cross-border studies on physical and mental health in different social groups.

## Figures and Tables

**Table 1 ijerph-19-13994-t001:** Characteristics of the study groups.

Variables		GROUP		Total (N = 779)
		ABNS (N = 374)	YKSUG (N = 405)	
		N (%)		
Sex	Male	105 (28.1)	97 (24.0)	202 (25.9)
	Female	269 (71.9)	308 (76.0)	577 (74.1)
Age	17 to 20 years	95 (25.4)	374 (92.3)	469 (60.2)
	21 to 30 years	210 (56.1)	31 (7.7)	241 (30.9)
	31 to 40 years	43 (11.5)	0 (0.0)	43 (5.5)
	over 40 years	26 (7.0)	0 (0.0)	26 (3.3)
Place of residence (during COVID-19 pandemic)	urban area	209 (55.9)	379 (93.6)	588 (75.5)
	rural area	165 (44.1)	26 (6.4)	191 (24.5)
Self-isolation due to COVID-19	yes	105 (27.0)	145 (35.8)	246 (31.6)
	no	273 (73.0)	260 (64.2)	533 (68.4)
Quarantine due to COVID-19	yes	150 (40.1)	83 (20.5)	233 (29.9)
	no	224 (59.9)	322 (79.5)	546 (70.1)
Vaccination against COVID-19	yes, a one-dose vaccine	46 (12.3)	103 (25.4)	149 (19.1)
	yes, two doses	162 (43.3)	126 (31.1)	288 (37.0)
	yes, three doses	64 (17.1)	1 (0.2)	65 (8.3)
	no	68 (18.2)	152 (37.5)	220 (28.2)
	I do not want to reply	34 (9.1)	23 (5.7)	57 (7.3)
Diagnosed with SARS-CoV-2	yes	101 (27.0)	145 (35.8)	246 (31.6)
	no	273 (73.0)	260 (64.2)	533 (68.4)

**Table 2 ijerph-19-13994-t002:** Homogeneity of groups under study.

Variable	Coefficient	Value	Chi-Squared	*p*
Sex	Phi	0.047	1.722	0.189
Age	V Kramer	0.687	367.270	0.001 *
Place of residence (during COVID-19 pandemic)	Phi	−0.438	149.310	0.001 *
Self-isolation due to COVID-19	Phi	−0.249	48.183	0.001 *
Quarantine due to COVID-19	Phi	0.185	26.578	0.001 *
Vaccination against COVID-19	V Kramer	0.393	120.520	0.001 *
Diagnosed with SARS-CoV-2	Phi	−0.185	26.561	0.001 *

* statistically significant differences at *p* < 0.05.

**Table 3 ijerph-19-13994-t003:** Physical activity levels in students from Poland and Belarus (IPAQ).

PA Levels	Group		Total (N = 779)	V Kramer	Chi-Squared	*p*
	ABNS (N = 375)	YKSUG (N = 405)				
	N (%)					
Low	56 (15.0)	12 (3.0)	68 (8.7)	0.228	40.41	0.001 *
Moderate	128 (34.2)	192 (47.4)	320 (41.1)			
High	190 (50.8)	201 (49.6)	391 (50.2)			

* statistically significant differences at *p* < 0.05.

**Table 4 ijerph-19-13994-t004:** Differences in test results between groups ABNS and YKSUG.

VARIABLES	GROUP	M (±SD)	Mean Equality Test		Effect Size		
			t	*p*	Cohen’s d	Hedges’ g	Glass’s Delta
SWLS	ABNS	20.78 (±5.49)	14.51	0.001 *	1.040	1.039	0.997
	YKSUG	14.76 (±6.03)					
Mini-COPE							
Active coping	ABNS	2.13 (±0.60)	5.96	0.001 *	0.423	0.422	0.375
	YKSUG	1.83 (±0.81)					
Planning	ABNS	2.01 (±0.66)	3.30	0.001 *	0.235	0.235	0.219
	YKSUG	1.84 (±0.77)					
Positive revaluation	ABNS	1.70 (±0.73)	−0.15	0.883	−0.011	−0.011	−0.010
	YKSUG	1.71 (±0.80)					
Acceptance	ABNS	1.89 (±0.70)	4.87	0.001 *	0.348	0.348	0.335
	YKSUG	1.63 (±0.76)					
Humourous approach	ABNS	1.10 (±0.73)	−8.96	0.001 *	−0.638	−0.637	−0.586
	YKSUG	1.63 (±0.89)					
Turning to religion	ABNS	1.09 (±0.99)	0.13	0.894	0.010	0.010	0.010
	YKSUG	1.08 (±0.94)					
Seeking emotional support	ABNS	1.86 (±0.82)	−0.14	0.891	−0.010	−0.010	−0.010
	YKSUG	1.87 (±0.82)					
Seeking instrumental support	ABNS	1.82 (±0.81)	1.70	0.090	0.122	0.122	0.124
	YKSUG	1.72 (±0.78)					
Dealing with other things	ABNS	1.80 (±0.69)	6.55	0.001 *	0.469	0.469	0.457
	YKSUG	1.46 (±0.73)					
Denial	ABNS	0.88 (±0.73)	−3.36	0.001 *	−0.241	−0.241	−0.235
	YKSUG	1.06 (±0.76)					
Giving vent to one’s feelings	ABNS	1.43 (±0.66)	−1.14	0.255	−0.082	−0.082	−0.081
	YKSUG	1.49 (±0.68)					
Taking psychoactive substances	ABNS	0.48 (±0.71)	−3.21	0.001 *	−0.230	−0.229	−0.220
	YKSUG	0.65 (±0.78)					
Doing nothing	ABNS	0.76 (±0.63)	−2.71	0.007 *	−0.194	−0.194	−0.186
	YKSUG	0.89 (±0.69)					
Self-accusation	ABNS	1.32 (±0.82)	−1.71	0.087	−0.123	−0.123	−0.120
	YKSUG	1.43 (±0.86)					
Mini-COPE INTEGRAL STRATEGIES							
Active coping	ABNS	1.95 (±0.51)	3.59	0.001 *	0.255	0.255	0.227
	YKSUG	1.79 (±0.68)					
Helplessness	ABNS	0.85 (±0.53)	−3.31	0.001 *	−0.238	−0.237	−0.225
	YKSUG	0.99 (±0.60)					
Seeking support	ABNS	1.84 (±0.76)	0.82	0.414	0.059	0.058	0.059
	YKSUG	1.79 (±0.76)					
Avoidance behaviours	ABNS	1.37 (±0.47)	0.90	0.368	0.064	0.064	0.059
	YKSUG	1.34 (±0.56)					
GHQ							
GHQ A somatic symptoms	ABNS	14.16 (±4.06)	−4.29	0.001 *	−0.305	−0.305	−0.281
	YKSUG	15.53 (±4.90)					
GHQ B anxiety and insomnia	ABNS	13.98 (±4.97)	−2.53	0.011 *	−0.181	−0.180	−0.169
	YKSUG	14.96 (±5.81)					
GHQ C social dysfunction	ABNS	13.97 (±13.98)	−1.39	0.164	−0.099	−0.099	−0.090
	YKSUG	14.31 (±3.73)					
GHQ D depression symptoms	ABNS	10.35 (±4.78)	−2.33	0.020 *	−0.167	−0.167	−0.170
	YKSUG	11.14 (±4.65)					
GHQ overall score	ABNS	52.45 (±13.42)	−3.37	0.001 *	−0.240	−0.240	−0.226
	YKSUG	55.93 (±15.40)					
IPAQ [MET-min/week]	ABNS	4777.24 (±5197.34)	4.936	0.001 *	0.364	0.363	0.636
	YKSUG	3339.33 (±2259.33)					

* statistically significant differences at *p* < 0.05.

**Table 5 ijerph-19-13994-t005:** Differences in test results between groups ABNS and YKSUG-non-parametric test.

VARIABLES	GROUP	Mean Rank	U Mann-Whitney	*p*
IPAQ [MET-min/week]	ABNS	406.78	69,460.5	0.046 *
	YKSUG	374.51		

* statistically significant differences at *p* < 0.05.

**Table 6 ijerph-19-13994-t006:** Correlations between the evaluation of mental health state of the respondents and their PA levels and satisfaction with life during COVID-19 pandemic (Spearman’s rho).

GROUP	TEST	IPAQ	SWLS
ABNS (N = 374)	GHQ A—somatic symptoms	−0.115 *	−0.209 ***
	GHQ B—anxiety and insomnia	0.014	−0.297 ***
	GHQ C—social dysfunction	−0.012	−0.244 ***
	GHQ D—depression symptoms	−0.003	−0.436 ***
	GHQ—overall score	−0.026	−0.371 ***
YKSUG (N = 405)	GHQ A—somatic symptoms	−0.063	0.216 ***
	GHQ B—anxiety and insomnia	−0.057	0.284 ***
	GHQ C—social dysfunction	−0.095	0.266 ***
	GHQ D—depression symptoms	−0.013	0.258 ***
	GHQ—overall score	−0.068	0.313 ***

* significant correlation at 0.05 (two-way); *** significant correlation at 0.001 (two-way).

**Table 7 ijerph-19-13994-t007:** Correlations between satisfaction with life and PA in the study participants during COVID-19 pandemic (Spearman’s rho).

TEST	GROUP	IPAQ
SWLS	ABNS (N = 374)	0.044
	YKSUG (N = 405)	0.028

**Table 8 ijerph-19-13994-t008:** Correlations between coping with stress and physical activity, mental health and life satisfaction during COVID-19 pandemic (Spearman’s rho).

VARIABLES	IPAQ	SWLS	GHQ A	GHQ B	GHQ C	GHQ D	GHQ General
ABNS (N = 374)							
Active coping	0.102 *	0.207 ***	−0.117 *	−0.087	−0.120 *	−0.272 ***	−0.174 ***
Planning	0.083	0.152 **	−0.084	−0.058	−0.101	−0.199 ***	−0.127 *
Positive revaluation	0.040	0.222 ***	−0.171 ***	−0.059	−0.147 **	−0.265 ***	−0.196 ***
Acceptance	−0.010	0.098	−0.099	−0.049	−0.043	−0.086	−0.085
Sense of humor	−0.002	0.050	−0.099	−0.089	−0.058	−0.040	−0.108*
Turning to religion	−0.053	0.154 **	−0.044	−0.003	−0.022	−0.102*	−0.060
Seeking emotional support	−0.062	0.224 ***	−0.010	−0.012	−0.062	−0.196 ***	−0.085
Seeking instrumental support	−0.081	0.150 **	0.058	0.064	−0.064	−0.158 **	−0.011
Dealing with other things	0.038	0.000	−0.011	0.120 *	0.012	0.019	0.043
Denial	−0.004	−0.071	0.178 ***	0.215 ***	−0.036	0.070	0.162 **
Giving vent to one’s feelings	−0.032	−0.089	0.164 ***	0.294 ***	0.064	0.146 **	0.231 ***
Taking psychoactive substances	−0.017	−0.151 **	0.121 *	0.195 ***	0.062	0.245 ***	0.194 ***
Doing nothing	−0.090	−0.244 ***	0.125 *	0.156 **	0.136 **	0.277 ***	0.198 ***
Self-accusation	0.005	−0.319 ***	0.237 ***	0.393 ***	0.225 ***	0.477 ***	0.427 ***
INTEGRAL STRATEGIES							
Active coping	0.110 *	0.253 ***	−0.160 **	−0.087	−0.162 **	−0.323 ***	−0.217 ***
Helplessness	−0.030	−0.322 ***	0.213 ***	0.357 ***	0.200 ***	0.472 ***	0.386 ***
Seeking support	−0.080	0.198 ***	0.025	0.026	−0.073	−0.191 ***	−0.054
Avoidance behaviours	−0.013	−0.091	0.173 ***	0.312 ***	0.030	0.124 *	0.224 ***
YKSUG (N = 405)							
Active coping	0.053	−0.053	−0.051	−0.093	−0.129 **	−0.160 ***	−0.152 **
Planning	0.039	−0.039	−0.007	−0.015	−0.146 **	−0.092	−0.082
Positive revaluation	0.064	−0.069	0.032	−0.046	−0.041	−0.146 **	−0.074
Acceptance	0.010	0.085	0.087	0.006	0.031	0.027	0.032
Sense of humor	0.054	0.107 *	0.084	0.050	0.074	0.068	0.076
Turning to religion	0.017	−0.019	0.155 **	0.163 ***	0.064	0.100 *	0.163 ***
Seeking emotional support	0.019	−0.030	0.002	−0.034	−0.057	−0.102 *	−0.072
Seeking instrumental support	0.024	0.014	0.036	0.012	0.011	−0.054	−0.010
Dealing with other things	0.052	0.133 **	0.146 **	0.167 ***	0.125 *	0.122 *	0.173 ***
Denial	−0.011	0.186 ***	0.162 ***	0.190 ***	0.125 *	0.237 ***	0.210 ***
Giving vent to one’s feelings	−0.010	0.099 *	0.195 ***	0.153 **	0.146 **	0.102 *	0.174 ***
Taking psychoactive substances	0.039	0.086	0.181 ***	0.229 ***	0.141**	0.280 ***	0.268 ***
Doing nothing	−0.043	0.191 ***	0.216 ***	0.231 ***	0.233 ***	0.311 ***	0.305 ***
Self-accusation	−0.023	0.326 ***	0.286 ***	0.355 ***	0.256 ***	0.357 ***	0.385 ***
INTEGRAL STRATEGIES							
Active coping	0.066	−0.067	−0.027	−0.069	−0.129**	−0.168***	−0.134**
Helplessness	−0.006	0.286 ***	0.316 ***	0.374 ***	0.291 ***	0.427 ***	0.441 ***
Seeking support	0.022	−0.010	0.013	−0.018	−0.028	−0.086	−0.050
Avoidance behaviours	0.002	0.177 ***	0.220 ***	0.232 ***	0.173 ***	0.216 ***	0.252 ***

* significant correlation at 0.05 (two-way); ** significant correlation at 0.01 (two-way); *** significant correlation 0.001 (two-way).

## Data Availability

Not applicable.

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
