# Peer review of "Physical Activity and Mental Health of Medical Students from Poland and Belarus-Countries with Different Restrictive Approaches during the COVID-19 Pandemic"

_ijerph, 2022, doi:10.3390/ijerph192113994_

Round 1

Reviewer 1 Report (Previous Reviewer 3)

The authors have improved their manuscript. 

Reviewer 2 Report (Previous Reviewer 2)

I have no comment.

Reviewer 3 Report (Previous Reviewer 1)

Dear. Authors,

I think you have answered my questions adequately, but further research needs to be accumulated. Looking forward to it.

Best,

This manuscript is a resubmission of an earlier submission. The following is a list of the peer review reports and author responses from that submission.

Round 1

Reviewer 1 Report

Dear. author, 

I would like to respectfully submit that you are providing very valuable data. However, as a research paper, I have some questions.

having some comments below.

1. overall, you need to change the position where you make sentence line spacing. Reduce unnecessary line spacing.

2. please provide details of the survey methodology. Details such as whether it was a web-based survey or a face-to-face questionnaire were distributed are not clear.

3. Tables should be revised to make it more readable to the reader. The shaded area is out of alignment and difficult to read.

4. The most important question is that you are conducting a comparison between the two organizations. Please explain in more detail why you need to compare these two organizations. Also, you have not made comparisons by gender or age group. You also states that "it is necessary to implement a prevention program", but in order to implement a prevention program, a comparison needs to be made using graphic information, even by age and gender.

Hope you would answer the above questions I have.

Best,

Author Response

Dear Reviewer,
Thank you for your valuable comments - we have tried to address each point

Yours sincerely

Reviewer 2 Report

The study aimed to assess mental health status and PA levels among students from countries in which different approaches to these issues were adopted during the SARS-CoV-2 virus pandemic. I add several comments below.

1.      The authors indicated the limitations that previous studies are more focus on correlations of physical activity and mental health with constant demographic variables. However, what’s the dynamically changing situation of the pandemic the study tried to stand out?

2.      Line 100, “...satisfaction with life during the COVID-19 pandemic was negatively correlated with time devoted to exercise, which may indicate that physically active individuals are more prone to well-being issues during the lockdown.” The statement is questionable.

3.      The duration and waves of Covid pandemic in Poland and Belarus needed to be demonstrated in detail, because those are the critical point of study design and are not in common among countries.

4.      The process of participants recruitment needs to be described.

5.      Table 1 showed some different characteristics between students in the two colleges. To avoid confounders, instead of Chi square test or t-test barely, a multi-variated analysis is suggested as a main statistics method.

6.      Some illustrations for study results are questionable. For example, Line 250, the more satisfied with life these students were, the fewer disorders the students experienced in YKSUG. I find there are positive correlations in Table 4, however.

7.      The results showed the associations in PA, satisfaction in life and mental health are not in common for the students of the two colleges. I consider it is hard to conclude solidly the correlation the study attempts to figure out.

8.      The authors concluded the differences of mental health between the two colleges may be caused by the different approaches in the two countries. However, what’s the difference of approaches in counties during the pandemic cannot be found. How can they prove the differences were caused by the approaches, not individual characteristics?

9.      I suggest the authors can reconstructed the abstract and the main text to lift the readability.

Author Response

(The authors gave the same response as above.)

Reviewer 3 Report

These are my comments on the article “Physical Activity and Mental Health of Medical Students from Poland and Belarus in the Context of the Dynamically Changing Situation of the COVID-19 Pandemic”. 

This manuscript aims to evaluate the pandemic effects on mental health and physical wellness of college students. The novelty lies on the comparison from different countries and methods. 

Introduction seems like discussion in several paragraphs. Authors should be more concise and paragraphs should be shortened.  

Materials and Methods are adequately described. 

Results are well presented. It seems that authors managed well their data and constructed interesting tables. Lines 269-319 are marginal discussion. Authors should be more direct to the results description and possible correlations should be presented on discussion (with or without comparisons to other studies). 

Several lines in discussion were mentioned in the results and should not be presented again, expect from commendatory sections with similar studies. 

I agree with the last statement of the conclusions “it seems necessary for universities to monitor physical and mental health state of students and to implement prevention programmes.” Indeed, as an initial measure, the Aristotle University of Thessaloniki (GREECE) has established a 24-h-communication line for members who seek counseling and psychological support. Records so far were in alignment with the evidence presented in this study. The cases of the University community members who seek consultation and psychological support has quintupled during the las two years. Therefore, it IS necessary for universities to monitor physical and mental health state…

 Authors should include the abovementioned discussion by the end of the discussion section alongside with the reference: https://pubmed.ncbi.nlm.nih.gov/35893354/  

All in all, this is an interesting paper, publishable after the above significant revisions. 

Author Response

(The authors gave the same response as above.)

Round 2

Reviewer 1 Report

Dear. Author,

I would like to suggest one comment. I think that a discussion based on demographic data is always necessary. You don't think it is necessary in this study because of the different objectives, but many of the articles in this journal have reported different effects on mental health depending on grade level, age, and other factors. I hope you can discuss the limitations of your research and future perspectives.

Best,

Reviewer 2 Report

The revised version has been amended better than the original one. However, there are severe study limitations that cannot be neglected still. The study sought to determine PA levels, satisfaction with life and mental health indicators in students from Poland and Belarus in the context of different approaches to the COVID-19 pandemic. The authors claimed that during the first wave of the pandemic, no decision was made to close the borders in Belarus. Neither schools nor shops were closed, and no sporting events were suspended. Generally, the first wave of pandemic is considered occurred in 2020 in most European countries. What’s going on the measures in the two countries in 2022 can’t be caught in the study. The study was carried out in April and May 2022, which make the findings less link to the different measures in the two countries during the pandemic. Moreover, the other factors such as students’ sociodemographic characteristics may play a main role on the effects of satisfaction with life and mental health, so the analysis with those factors should not be ignored. Additionally, the study did not reveal correlations between PA levels and depression symptoms, anxiety and insomnia, and life satisfaction. The findings are different from the general knowledge evidenced by most past studies and lack of contribution in science.